# Public Administration and Values Oriented to Sustainability: A Systematic Approach to the Literature

Isabel Marques [1], João Leitão [2,3,4,5,*] , Alba Carvalho [1] and Dina Pereira [3,4]

[1] Department of Management and Economics, University of Beira Interior (UBI), 6200-209 Covilhã, Portugal; isabel.marques@ubi.pt (I.M.); albakattarine@msn.com (A.C.)
[2] Faculty of Social and Human Sciences, NECE–Research Center in Business Sciences, University of Beira Interior, 6200-001 Covilhã, Portugal
[3] Research Center in Business Sciences (NECE), University of Beira Interior, 6200-209 Covilhã, Portugal; dina.pereira@tecnico.ulisboa.pt
[4] Centre for Management Studies of Instituto Superior Técnico (CEG-IST), University of Lisbon, 1049-001 Lisboa, Portugal
[5] Instituto de Ciências Sociais, ICS, University of Lisbon, 1600-189 Lisboa, Portugal
* Correspondence: jleitao@ubi.pt; Tel.: +351-275-319-853

**Abstract:** Values guide actions and judgements, form the basis of attitudinal and behavioral processes, and have an impact on leaders' decision-making, contributing to more sustainable performance. Through a bibliometric study and content analysis, 2038 articles were selected from Scopus, from the period 1994–2021, presenting global research tendencies on the subject of values, public administration, and sustainability. The results indicate that *Sustainability* is the most productive journal, the main research category is in social sciences, the most productive institution is the University of Queensland, the location with the most publications and research collaborations is the USA, and the authors with the greatest number of articles are Chung, from Chung-Ang University; García-Sánchez, from the University of Salamanca; and Pérez, from the University of Cantabria. Analysis of keywords shows that the most relevant are "sustainability", "CSR", "sustainable development", "innovation", and "leadership". Time analysis of keywords reveals a tendency for lines of research in the social and work area. The results also provide data about the framing of studies in sustainability pillars and the types of values referred to and indicate the main areas of public administration studied. Finally, a future research agenda is proposed.

**Keywords:** public administration; public sector; values; sustainability





## 1. Introduction

The main motivation of this systematic literature review is related to the scarcity of literature reviews that deal with the problem of public administration and values oriented towards sustainability. In this vein, it is necessary to map political, cultural, ethical, moral, aesthetic, ecological, vital, spiritual, and religious values to understand the different ways in which organizational and individual values have been addressed in the literature, as well as the contributions of skills, managerial techniques, and moral competence having a behavioral effect, which allows for the achievement of a balanced exercise of sustainability. In this sense, the administration of public institutions requires the adoption of ethical principles and values oriented towards the effective and efficient use of public resources with the ultimate aim of contributing to increased social well-being. Ethics and values, although often used as synonyms, do not have the same meaning, with ethics being the mental process that comes into action when the individual is deciding between right and wrong, or assessing two rights, where the process of appropriate decision-making principles is established for when different values enter into conflict [1]. For the same authors [1], having values represents understanding the importance of ethical processes for decision-making; being ethical helps in choosing the correct values.

From an organizational perspective, ethics should be understood as a horizon of shared values, where organizational practices are directed and transformed simultaneously, creating an important ethical meaning [2]. In organizations, true values are noted in leaders' decisions, in the way people are rewarded and promoted, in the assessment methodology, and in corporate practices [3], and they can influence various aspects, such as employees' feelings towards the organization [4] and human well-being [5]. Bettinger [6] states that a robust corporate culture, where values are rooted in the organization, is one of the key factors contributing to long-term, sustained high performance, as values allow greater work engagement, and consequently, greater prosperity [7].

In this context, value-based management is a process of rooting values continuously at the heart of organizations in order to form a true culture of values. Dolan and Garcia [8] indicate that value-based management is a strategic management tool with the triple usefulness of simplifying, guiding, and developing employees' behavior. However, despite efforts to create value for these and other stakeholders, organizations have not managed to implement, fully and effectively, their sustainability policies [9].

The notion of values is especially appropriate in public service [10], considering that this should be permeated by values that benefit citizens collectively [11], covering aspects of integrity, legality, and participation related to good governance [12]. Recently, conflicts of values in the circumstance of the modernization of public services have been shown, particularly regarding the principles of reform concerning the paradigm of New Public Management (NPM), which aims to ensure greater efficiency and effectiveness of public services in countries belonging to the Organization for Economic Cooperation and Development (OECD) [13]. Public service is substantially different from the private sector, considering that rather than dealing with a market environment and with a limited number of stakeholders, public service relates to a political environment involving various individuals and external organizations [14]. In addition, government objectives and restrictions placed by political authorities ensure that private sector practices cannot be indiscriminately transferred, partly due to the different operational environment, which seems to have an impact on organizational culture and to be a challenge in public service.

Concerning public management specifically, although the subject is relevant, with a significant number of studies addressing organizational values, few studies focus on the dynamics inherent to the concept of values, allowing theoretical constructions and bibliographic measurements in the context of public management, which is also related to organizations' sustainability.

In this context of analysis, it is considered relevant to verify the typology of the values adopted in the administration of public institutions, which, combined with managers' technical skills and moral competence, can ensure the ultimate goal of sustainability.

This systematic approach contributes to the literature by mapping the main streams, areas, contributions, contributors, institutions, and locations associated with the topic of values attributed to sustainable public administration, providing a future research agenda grounded on the most important clusters found in the set of reference studies selected.

The current systematic literature review is organized as follows. Firstly, it provides an overview of the main literature streams and the conceptual framework in use. Secondly, it presents the materials and methods. Thirdly, the results and discussion are presented, based on bibliometric analysis and qualitative content analysis. Fourthly, the conclusion and avenues for future research are provided.

## 2. Literature Review and Conceptual Framework

### 2.1. Axiology and Organisational Values

Axiology concerns the study of values in a wide-ranging way, expanding their meaning and articulating economic, ethical, aesthetic, and logical questions that are traditionally considered separately. For Modin [15], Nietzsche is the father of axiology, although Lotze was the proposer. This is because Nietzsche, with his criticisms, sought to break down all the absolute values of logic (truth), morality (virtue), metaphysics (being) and religion

(God), pointing to their decadence and alienation. Secondly, he proposed the dynamism of the value of life, a life that accepts in itself all its expressions. For McArdle, Hurrell, and Muñoz Martinez [16], axiology allows an understanding of the influences of beliefs and values on life experiences, human actions, and perceptions.

Until the mid-1970s, research was formed of empirical studies, but these were limited by groups' own particular construction and the methods adopted, which did not allow systematic comparison between the studies made. Research carried out in 1973 [17] harmonized the relation between systematic research of the theory of values and established the connection between values and behavior, where by recognizing their values, individuals should predict their behavior. Other authors [18] consider values as being principles acting on behaviors, going far beyond specific situations and directing behaviors, ordering them according to their relevance. A more contemporary definition [19] summarizes values as lasting beliefs that serve as a reference for action and vary very little according to the circumstances. In turn, Schwartz [20], developing the theory of universal basic human values, presents the fundamental characteristics for values that form a system of priorities characterizing individuals and guiding actions. For this author [21], individual values differ from cultural ones, considering that the individual's axiological priorities are the result of shared culture and unique personal experience, whereas cultural values help society to mold contingencies to which people must adjust. Frunză [22] proposes that values can be used as resources to guide actions, situations, and states in organizations.

In recent decades, values have been defined as the classification of collective principles orienting action, or how a collective tries to act, forming a consensus that a social or organizational group judges to be relevant to achieve objectives and collective well-being [22]. Frunză [22] states that the idea of the relativization of values has spread in recent times, allowing the formation of new hierarchies of values considering individual or organizational needs. The author also mentions that the organization should be seen not just as a place of work, but also as part of employees' development and personal fulfilment.

## 2.2. Public Administration and Competing Values

Authors discuss that values between private and public organizations differ in practice, and this is an empirical question, considering that they are shown through attitudes, preferences, decision-making, and actions [23]. With globalization, public organizations have come under pressure to find a common governance system. The consequence of this is the increasing similarity between the values and principles of public administration worldwide [24]. Oldenhof, Postma, and Putters [25] conclude that public management is characterized by multiple conflicting values, such as impasses between efficiency and equity, efficiency and democratic legitimacy, and equity and freedom, among others. The study by Van der Wal, Graaf, and Lasthuizen [23] follows in this direction, finding that although public managers consider values such as legality, impartiality, and incorruptibility as the most important in the sector, many private sector values, such as expertise and effectiveness, are also indicated by these professionals. In this context, Villoria [26] points out that the managers of public organizations face four types of value conflicts: (i) between political and organizational values; (ii) between organizational and social values; (iii) between organizational and economic values; and (iv) between economic values themselves. These conflicts require of managers solid decision-making that is aware of the consequences, marked by technical skills but also by moral competence [27].

Renshaw, Parry, and Dickmann [28] point out that many studies address the term "organizational values" without a proper definition, probably due to the difficulty of conceptualizing it, bearing in mind that values are often positioned as constructs at the individual level [29]. However, it should be noted that organizational values are important components of the organizational culture [30] and principles that are responsible for the successful management of organizations [31]. Concerning individual values, these can be defined as the internalized beliefs held by individuals about the way they should

behave [32], according to their personal experiences [33], culture, and the social system where they are inserted [31,33].

Thus, it is also important to note that there is a tension between both types of values. Public employees are exposed to conflicting demands, which they must meet [34]. This conflict stems from the need to respond to citizens, align their decisions with the interests of co-workers [35], and adapt their preferences, functions, and identities to the organization in which they are located. At the same time, public institutions need to meet demands, presenting the best results and using fewer resources, and have the challenge of integrating these employees into the organization and its standards. This adjustment process between the parties corresponds to so-called organizational socialization [36].

Individuals who are part of public organizations are often frustrated with their restrictions and ambiguities [37] and with the organizational ownership of values in the public service [38]. Considering the importance of congruence between organizational and individual values, in terms of individuals' attitudes and behavior [39,40], to reach better individual performance [39–41] and to ensure the organization's success [41], the key role played by the leaders to encourage this alignment of values is considered fundamental. In this ambit, transformational leadership is a powerful a tool that can be used by managers to promote an oriented process for alignment and congruence of values in the public service [41,42].

Public organizations come under double pressure: on one hand, competitiveness and the need to improve their economic position, service provision, and efficiency, and on the other, pressure to maintain traditional, historical management practices [43]. Therefore, there is discussion about the set of values that should guide public administration [44], with the consensus being that there are multiple value systems that are often in conflict, such as the dualities, for example, between efficiency and effectiveness, or impartiality and legality. Public organizations present the need for "renewal" in both political and administrative aspects in order to obtain the best strategies for their institutions to achieve their objectives, providing society with efficient services [45]. To do so, organizational culture is one of the key points in understanding human actions [46].

In this context, Peng and Pandey [47] mention the importance of autonomy as a driver of individuals internalizing public organizational values, which is in the public interest as these tend to help them face the conflicts experienced by individuals concerning the complex decisions of the actions of public administration management. In the context of New Public Management (NPM), civil servants are expected to perform their roles, duties, and tasks differently in responding to citizens' needs and demands. This is reflected in a call for a normative change in the organizational culture of the public sector, which should be accompanied by a similar change in the team's vision, perceptions, and will to adapt to that culture. Therefore, the change in values, at both the macro-organizational and micro-personal level, should be aligned to create a harmonious functioning of modern bureaucracy [48]. For the authors, new policies will be successful only when the individual is comfortable with the process of value transformation and the new organizational culture. Those values portray the need to improve internal and external processes of effective management, strengthening the relationship with the public and developing strategic thinking directed at clearer and measurable objectives, in the same ways as private companies operate [48]. As for the hierarchical aspect, as occurs with people, organizations differ, not so much through having different values, but by the degree of importance given to each of them, indicating that values represent guidelines for the organization's life, influencing its members' behavior [49], with shared values having important functions in the organizational context.

Zhong, Bao, and Huang [43] say that the values of work in public organizations have been subject to considerable study in recent decades, mainly after NPM. Management based on values is considered an important tool for human resource management [50], as it guides employees' attitudes and influences their performance at work [51].

### 2.3. Public Value and Organizations' Sustainability

Following the UNESCO (United Nations Educational, Scientific, and Cultural Organization) vision, sustainable development is the comprehensive paradigm of the United Nations, described by the 1987 Brundtland Commission Report as "development that meets the needs of the present without compromising the ability of future generations to meet their own needs" [52]. Sustainable development has four interconnected dimensions (i.e., society, environment, culture, and economy). Sustainability is a paradigm facing the future in which environmental, social, and economic considerations are balanced in the search for improving the quality of life, which is considered a long-term goal, while sustainable development refers to the various processes and paths to achieve it. In 2015, all member states of the United Nations adopted the Sustainable Development Goals containing 169 goals that countries seek to achieve by 2030. According to Clar et al. [53], the public sector faces some barriers to the adoption of policies that lead to sustainable development, such as lack of commitment, inadequate or unclear responsibilities, inadequate cooperation between political actors, insufficient financial and human resources, and lack of evidence or certainty in relation to global scenarios [53]. Promoting governance and providing better ways to link science to policymaking enables decisions to be based on good research that emphasizes trade-offs and multiple possibilities for action [54]. The dimensions of public value focus on organizational performance, presented as a benefit [55] and covering the dimensions of efficiency and effectiveness of public value and environmental sustainability, reflected in critical public value and understood positively by citizens in contexts of developing countries [56]. Some government projects have emerged to encourage public managers to embrace practices that arouse socio-environmental responsibility, while also approving responsible practices as a part of their long-term sustainability strategy [57]. Public administration is based on satisfying two needs: society demands creative, flexible treatments directed towards innovation, while economic tension and budget cuts force uses and models directed towards efficiency, competitiveness, and cost economy [58]. The new organizational system of public governance must integrate creativity, innovation, and flexibility to be able to achieve sustainability and public value.

Public value is related to a principle that must be continuous or a standard that must be met by public organizations when they regulate or provide their services, and it can be exteriorized through codes, ethics, norms, or principles [59]. Such values help organizational members to understand how they should act in that organization [60], serve as a link between civil servants' daily work and the general objectives of democratic governance [61], and are relevant for organizations, individuals, and societies. Attitudes and interpersonal interactions in organizational environments are affected by values and are seen as stimulating personal choices [62]. By taking with them their personal beliefs, choices, and actions into organizations, individuals count on them to make decisions [63], even if what forms the value(s) of public service differs from one country to another [64], considering that understanding the roles of values involves much questioning related to ethics in public administration [65]. For Hossain et al. [10], public service should serve the public interest as a way to ensure institutions' sustainability, and although some unethical and illegal practices persist, leading to public distrust, not all the behavior of civil servants is considered unethical, and some of this should be encouraged by stimulating practices to ensure such behavior is applied.

This reflection leads to the presentation of a conceptual structure (Figure 1, below) showing that different values, in both public administration and the sphere of private services, should be built and shared setting out from individuals, through the culture molded in organizations, and followed by top management and organizational members. Despite the existence of possible conflicts between those values, the behavioral effect arising from managers' technical competences and morals leads to organizations' social, economic, and environmental sustainability. The double bond "refers to the two feedback circuits that connect the observed effects of the action with strategies and values served by the strategies" ([66], p. 21). In this study, both organizational and individual values

are connected with technical skills at the level of competence of the managers who have a critical influence on the sustainability of public institutions. The combination of values, skills, and level of competence can stimulate new entrepreneurial practices that will lead to a sustainable pathway. Authentic leadership depends on the organizational context and an individual's positive psychological attitude, and it determines the self-awareness and self-regulated positive behavior of leaders and employees [67].

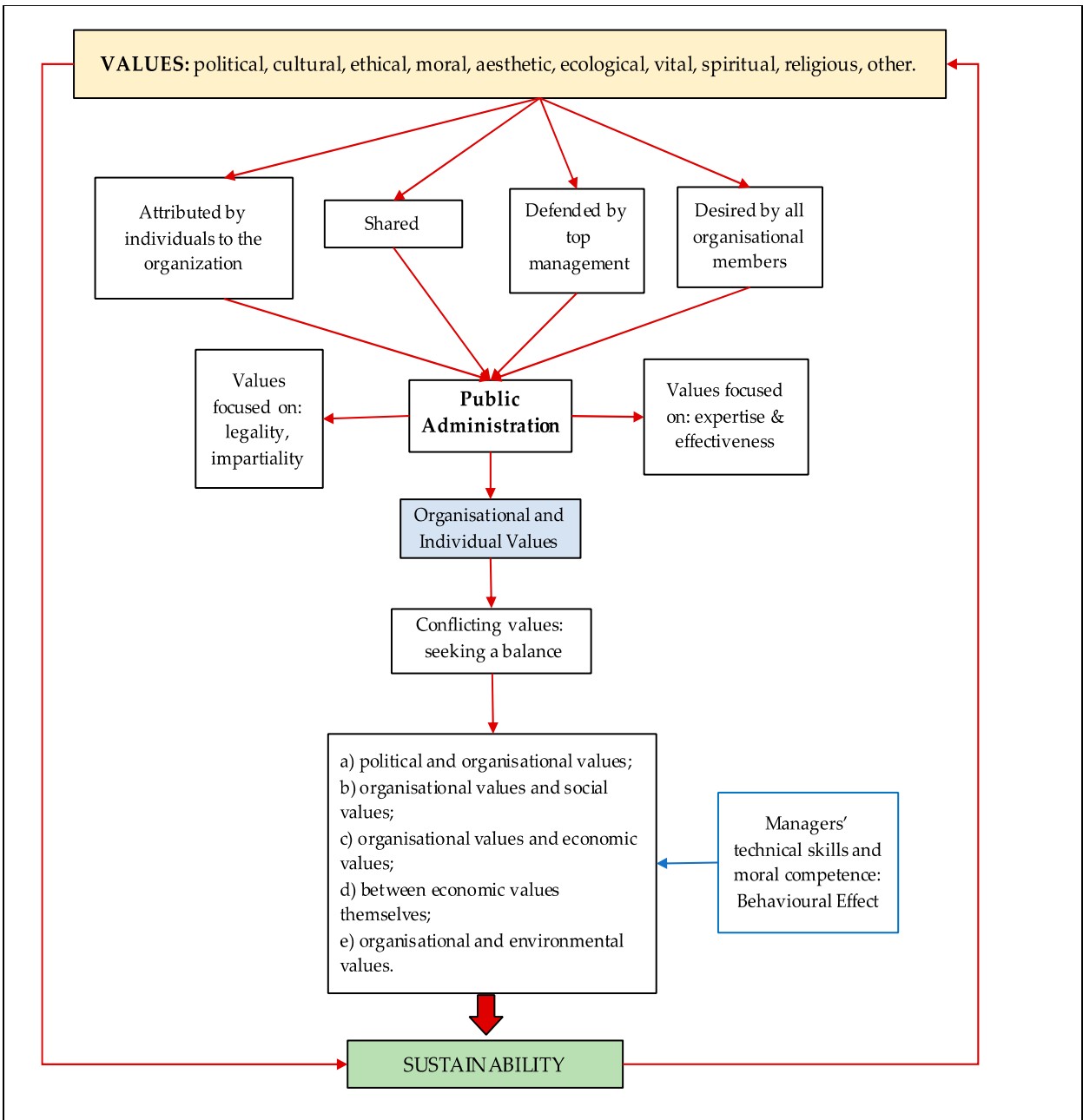

**Figure 1.** Conceptual structure of organizational values for institutions' sustainable reach. Source: Elaborated by the authors, adapted from Van der Wal, Graaf, and Lasthuizen [23]; Villoria [26]; and Ling, Wang, and Feng [27].

## 3. Materials and Methods

The research methods used were bibliometrics and content analysis of articles selected from the Scopus database (collected on October 8, 2020) due to its extent, reliability, and coverage [68]. The Scopus database was chosen due to its multidisciplinary nature and

large coverage. In addition, it is peer-reviewed, has daily updates, and has resources that assist the user in the searches carried out on the website, as well as creation lists for storing documents in the database during the search session, with structured searches by author and subject. The main advantages are: (i) inclusion of titles available in open access; (ii) wide coverage in terms of science and technology journals; (iii) tools for identifying authors; (iv) automatic generation of the h-index; (v) inclusion of more European content than Web of Science (WoS); and (vi) integration of more languages than English. An interesting feature is that although the Scopus database was not designed as a citation index, it includes citations from articles dating back to 1996.

From a number of options for choosing articles guiding the values of public administration, the key words of "public administration", "public sector", "values", and "sustainability" were used in the "All" box, adding to the search "and", which resulted in 9698 articles. Limiting the areas reduced this to 2038 articles, as presented in Figure 2.

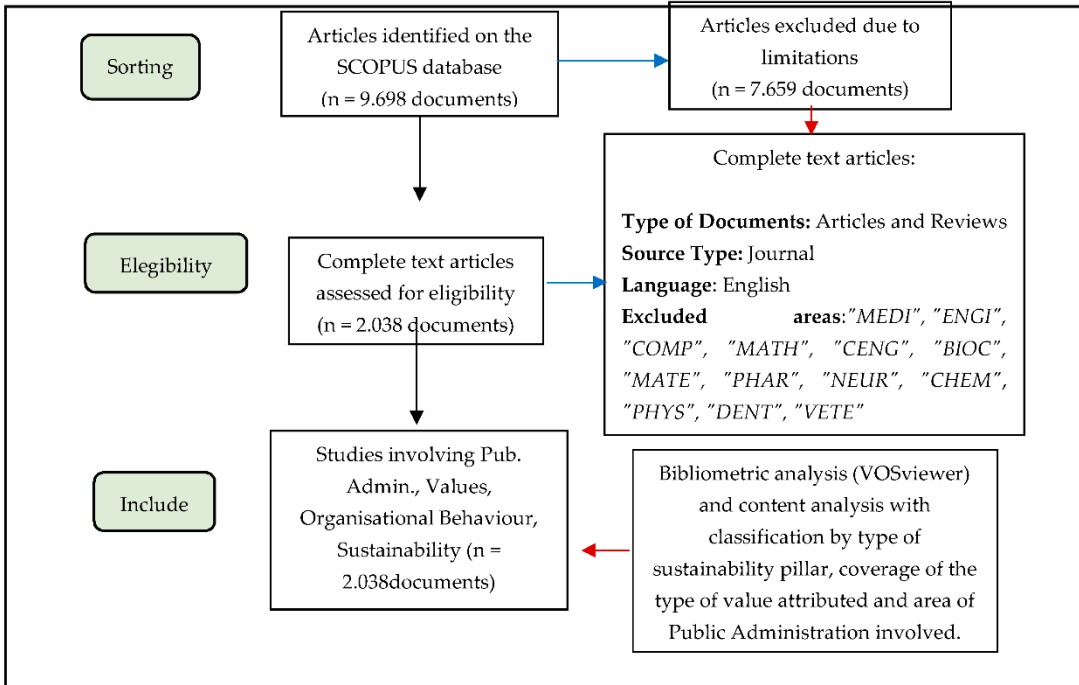

**Figure 2.** Method for selecting and including articles for analysis. Source: Own elaboration.

For the bibliometric analysis—which is the study of the measurement of scientific and technological progress and consists of quantitative assessment and analysis of comparisons of scientific activity, productivity, and progress [69]—VOSviewer (version 1.6.10., University of Leiden, Leiden, the Netherlands) was used to map and process the articles due to its reliability and suitability for bibliometric analyses. The relations between authors, institutions, and co-authors' locations were analyzed and interpreted based on the co-authorship of each study, and the relations between the keywords of all the documents was analyzed based on their co-occurrence [70]. Analysis of co-occurrence gives a graphic view of the interconnection of key terms in the documents [71]. Co-authorship analysis can reveal scientific collaboration and identify research groups [72].

For the qualitative stage, the data were studied using both an inductive and deductive method of content analysis [73,74]. The Abstract, Introduction and Results of the articles were read (i) to classify them as fitting one of the pillars of sustainability (environmental, economic, social) most associated with the subject of the article; (ii) to perform quantitative identification of the types of values found; and (iii) to identify associations between the values and sustainability pillars they are related to (political, ethical, ecological, moral, and others), (iv) the distribution of studies according to the major areas of public administration,

(v) the relation between the areas of public administration and the pillars of sustainability, and (vi) the relation between the areas of public administration with greatest emphasis and the values attributed to them. In this way, researchers, academics, managers, and others can benefit from the results arising from assessment in this area of research.

## 4. Results and Discussion

### 4.1. Bibliometric Analysis

4.1.1. Publications, Citations, and Areas of Research

Figure 3 shows the trend of evolution of publications on the topic studied. From the final selection (n = 2038), the articles were stratified according to the dates of publication, which cover the 1994–2020 period. From the total sum of 2038 articles identified, 1449 were published in the last 5 years, 2016 to 2020, which is 71.09% of all scientific production in these 28 years. This result reveals the growing interest and relevance of the topic.

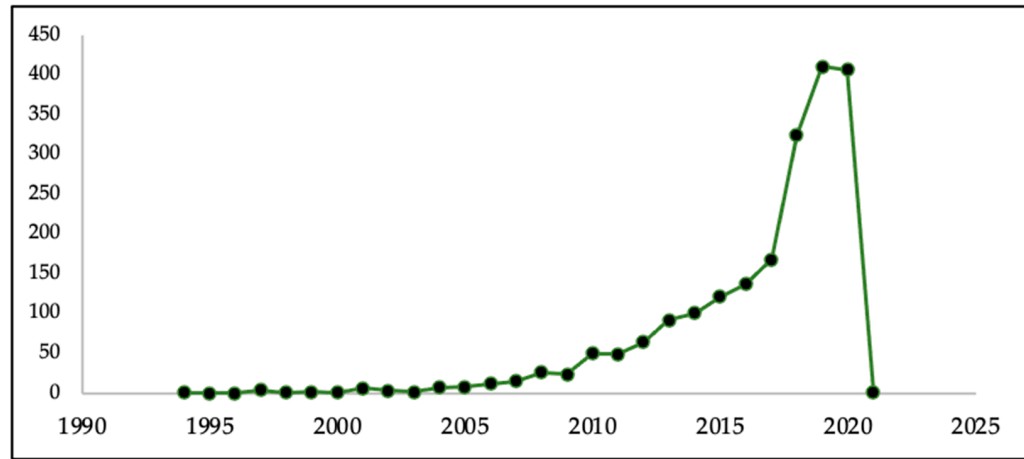

**Figure 3.** Number of articles included in the study, over 28 years. Source: Own elaboration.

Analyzing the evolution of citations (Figure 4), articles published before 2000, despite being few in number, had 13.758 citations, demonstrating that these articles have served as the basis to build scientific knowledge on the topic. In comparison, the most recent articles present a lower number of citations. This may be explained by the high number of articles, which has the effect of a greater distribution of citations among them.

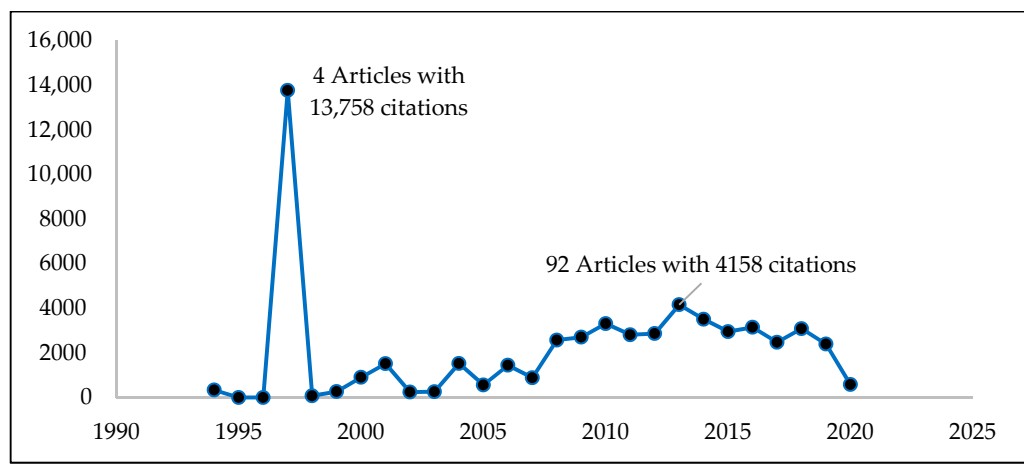

**Figure 4.** Evolution of the number of citations of articles per year. Source: Own elaboration.

According to the Scopus database, the journal clearly containing the most research in the area was *Sustainability Switzerland*, with 465 articles published (22.81% of the to-

tal), followed by *Journal of Business Ethics*, with 74 articles, and *Business Strategy and the Environment*, with 45 articles (Figure 5).

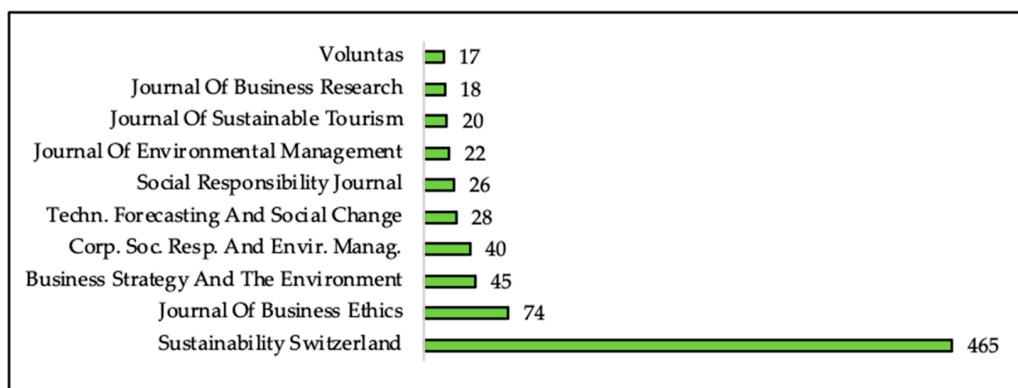

**Figure 5.** Main journals contributing to the theme. Source: Own elaboration.

The distribution of articles by area of research is presented in Figure 6. There is a concentration of articles in the areas of social sciences (58.97%), business management and accountability (51.07%), and environmental science (41.95%).

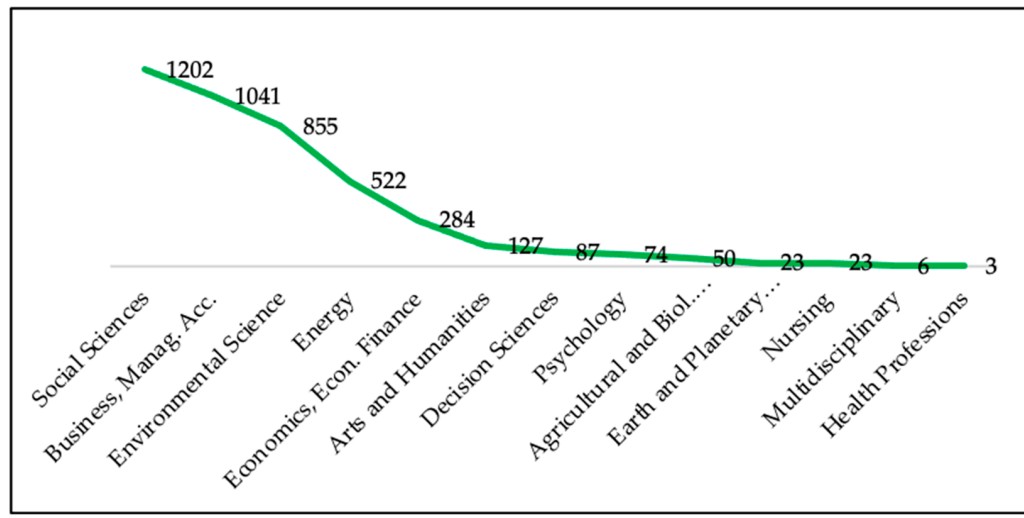

**Figure 6.** Main research areas. Source: Own elaboration.

According to the data extracted from Scopus, the most relevant articles in the area of social sciences address topics related to strategies for sustainability, educational administration, adaptation to environmental changes, tourism management, corporate sustainability, and other subjects. Regarding the area of business management and accountability, among the topics addressed are dynamic capabilities, strategic management, mindful consumption, corporate responsibility, and green supply chain management. Finally, in the area of environmental science, the most relevant articles include analysis of stakeholders, regional social-ecological systems, global climate change, and social license to operate.

### 4.1.2. Articles by Author, Institution, and Location

The 10 most productive authors on the subject of public administration and values oriented to sustainability were Chung, C.Y. (Chung-Ang University); García-Sánchez, I.M. (University of Salamanca); Pérez, A. (University of Cantabria); Barrutia, J.M. (University of the Basque Country); Echebarria, C. (University of the Basque Country); Gallego-Álvarez, I. (University of Salamanca); Gunasekaran, A. (California State University); Hickey, G.M.

(McGill University); Khan, M. (Abu Dhabi University); and López-Gamero, M.D. (University of Alicante). These results show a predominance of Spanish authors in studies on the subject (Table 1). In addition, Table 1 shows that these main authors published their most recent work in the last decade, confirming the current relevance of the topic. As for the keywords related to authors, a first group is related to organizations' social responsibility and their value and reputation, being formed by the words "corporate social responsibility (CSR)", "sustainable development", "corporate governance", "corporate and firm value", "stakeholders", "environmental management and policy", "reputation", and "reporting". A second group of words is related to local development and is formed of the words "local Agenda 21", "networking benefits", "local governments", and "sustainability". Finally, a third group, focusing on institutional characteristics and practices in public administration, is formed by the words "institutional theory", "sustainable supply chain", "big data", "dynamic capability", "innovation", "governance", and "public administration".

**Table 1.** Main authors and keywords.

| Author | Affiliation | Location | First Article * | Last Article * | N° of Publications * | Main Keywords |
|---|---|---|---|---|---|---|
| Chung, C.Y. | Chung-Ang University | Korea | 2018 | 2020 | 6 | CSR, corporate governance, sustainable development, corporate value, firm value |
| García-Sánchez, I.M. | University of Salamanca | Spain | 2014 | 2020 | 6 | Sustainable development, environmental policy, environmental management, CSR, stakeholder engagement |
| Pérez, A. | University of Cantabria | Spain | 2015 | 2020 | 6 | CSR, reporting, reputation, stakeholders |
| Barrutia, J.M. | University of the Basque Country | Spain | 2011 | 2016 | 5 | Local Agenda 21, networking benefits, local governments, sustainability |
| Echebarria, C. | University of the Basque Country | Spain | 2011 | 2016 | 5 | Local Agenda 21, networking benefits, local governments, sustainability |
| Gallego-Álvarez, I. | University of Salamanca | Spain | 2012 | 2020 | 5 | Environmental, CSR, sustainable development |
| Gunasekaran, A. | California State University | USA | 2019 | 2020 | 5 | Institutional theory, sustainable supply chain, big data, dynamic capability |
| Hickey, G.M. | McGill University | Canada | 2013 | 2018 | 5 | Innovation, governance, sustainable development, public administration |
| Khan, M. | Abu Dhabi University | United Arab Emirates | 2018 | 2020 | 5 | Analytical hierarchy process, sustainable development, social sustainability, Sustainability |
| López-Gamero, M.D. | University of Alicante | Spain | 2008 | 2016 | 5 | Competitiveness, environmental strategy, environmental management, hotels |

Legend: * = in this research topic. Source: Own elaboration.

Figure 7 shows the network or map of cooperation between authors publishing on public administration and values oriented to sustainability, based on co-authorship. The color of each cluster refers to the group of authors in producing articles, while the size of the circle is interpreted according to the number of contributions made by the author. Here, authors are associated in seven clusters. Cluster 1 (red) presents the collaboration between Chen, X.; He, Q.; Jiang, Y.; Sial, M. S.; Zhang, I.; Zhang, S.; and Zhang, W. Cluster 2 (green) is formed by Abbas, L.; Chen, Y.; Gursoy, D.; Hu, X.; Wang, Y.; Wu, X.; and Zhang, Q. Cluster 3 (dark blue) presents the collaboration between Ariza-Montes, A.; Han, H.; Hernández Perlines, F.; Kim, I.; and Lee, S. Cluster 4 (yellow) is formed by Amin, A.; Liu,

Y.; Wu, J.; Yang, I.; and Zhang, D. Cluster 5 (purple) presents the collaboration between Nguyen, N.; Tsai, S. -B.; Wang, J.; and Xu, H. Cluster 6 (light blue) is formed by Lu, Y.; Xu, Y.; Zhang, H.; and Zhang, Y. Finally, cluster 7 (orange) is formed by Kim, H.; Kim, J.; and Sun, Y.

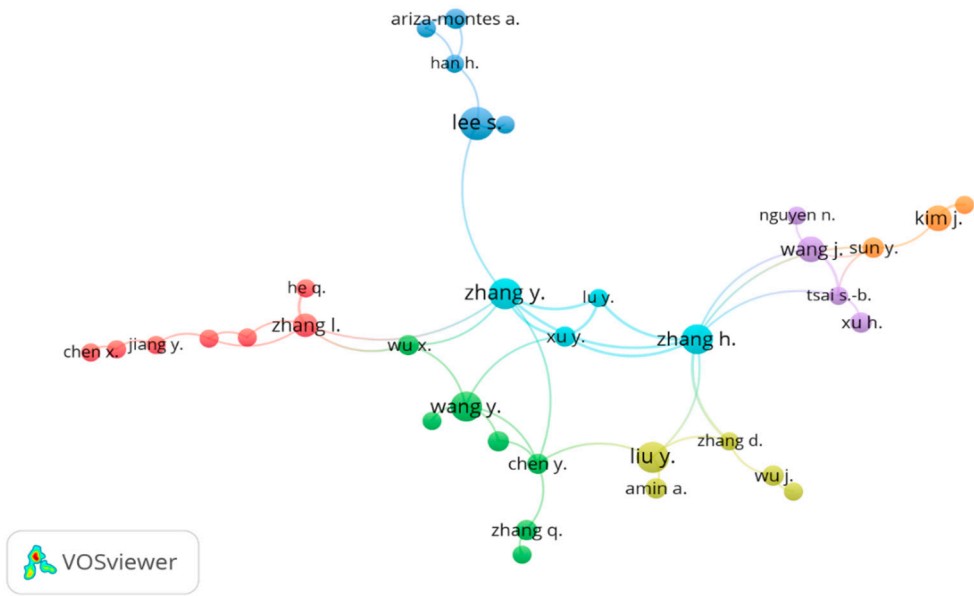

**Figure 7.** Cooperation network based on co-authorship between authors. Source: Own elaboration.

A predominance of Chinese authors is noted in cooperation networks. China has undergone various changes in its social system and administrative organizations, aiming to improve public service performance, following internationally accepted standards. This is a major challenge for the country, considering that traditionally its political system restricted citizens' participation [24].

Table 2 presents the 10 institutions that contributing most to scientific production on the subject studied. Standing out with 22 publications is the University of Queensland, located in Australia. It is followed by Hong Kong Polytechnic University, Wageningen University and Research, the University of Salamanca, the University of Granada, Bucharest Univ. Econ. Studies, Chung-Ang University, the University of Oxford, the University of Castilla-La Mancha, and the University of Waterloo. Therefore, of the 10 institutions contributing the most, three are Spanish.

The keywords related to the most prominent institution, the University of Queensland, were those related to the more social, non-profit-making aspect of organizations. In relation to the three Spanish universities, standing out are CSR, organizations' sustainable development, the hotel sector, and eco-innovation (Table 2).

The collaboration network between the main institutions publishing on public administration and values oriented to sustainability, based on co-authorship, is presented in Figure 8. Here, the colors represent the work groups publishing articles, while the size of each circle indicates the number of articles from each affiliation. Of the 4594 institutions, 159 of them present at least 2 articles, but they form 116 different clusters, with the biggest group consisting of only 5 connected institutions, namely Centrum Católica Graduate Business School, Pontifical Catholic University of Peru, Macau Polytechnic Institute, Macau University of Science and Technology, and Northeastern University. Given the high number of clusters formed, these results show a practically non-existent collaboration network between institutions, revealing scattered production of knowledge on this topic.

**Table 2.** Main institutions and keywords.

| Institution | Location | Nº of Publications * | Main Keywords |
|---|---|---|---|
| The University of Queensland | Australia | 22 | Social entrepreneurship, non-profit organizations, environmental management |
| Hong Kong Polytechnic University | China | 19 | Tourism workforce, CSR, environmental management |
| Wageningen University & Research | The Netherlands | 17 | Governance mechanisms, climate change adaptation, climate change decision |
| University of Salamanca | Spain | 17 | Environmental performance, stakeholder engagement, sustainability development, CSR |
| University of Granada | Spain | 17 | CSR, environmental strategies, sustainable development, information disclosure |
| Bucharest Univ. Econ. Studies | Romania | 17 | Sustainable business performance, investment decision, Business performance, sustainable entrepreneurship |
| Chung-Ang University | South Korea | 15 | Corporate value, CSR, firm value, corporate governance |
| University of Oxford | United Kingdom | 15 | Conflict resolution, GHG emissions reduction, Agenda 21, water–energy–food nexus |
| University of Castilla-La Mancha | Spain | 14 | Sustainable entrepreneurial orientation, family firms, hotel sector, eco-innovation |
| University of Waterloo | Canada | 14 | Agenda 21, stakeholders, tourism, visitor services, social network analysis |

Legend: * = concerning this research topic. Source: Own elaboration.

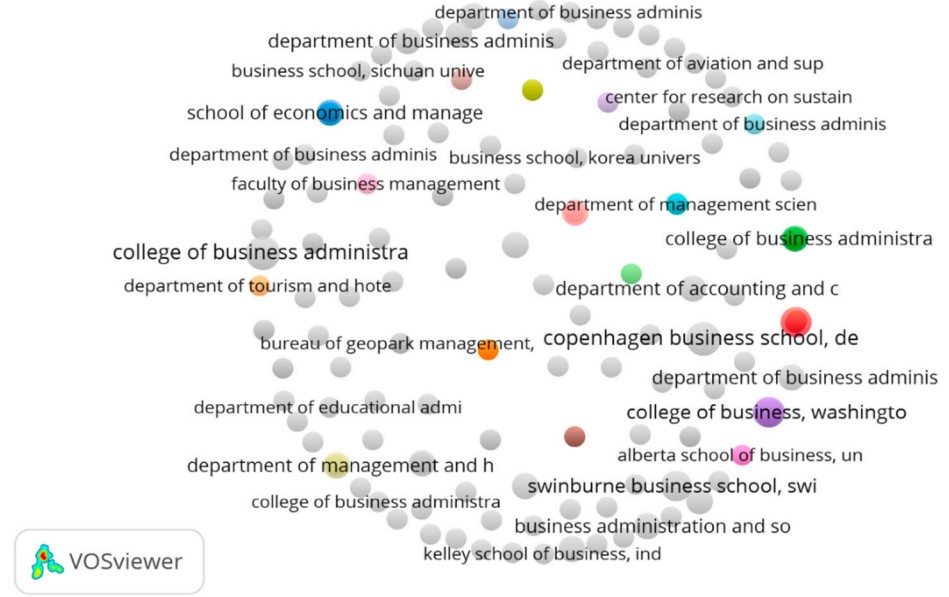

**Figure 8.** Network of cooperation between institutions based on co-authorship. Source: Own elaboration.

The total sample of articles originated in 116 different locations. The location with the greatest number of articles published on the subject is the USA (20.66%), followed by the United Kingdom (11.97%), Spain (10.2%), China (9.27%), Australia (8.04%), Canada (6.23%), Italy (6.08%) and South Korea (5.05%). The remaining locations do not exceed 5% of the number of articles.

The collaboration network between the main locations, considering co-authorship in the last 28 years, is presented in Figure 9. The colors represent the different clusters formed by groups of locations, while the size of the circle varies according to the number of items per location. In corresponding terms, the locations contributing to this area of research formed seven clusters.

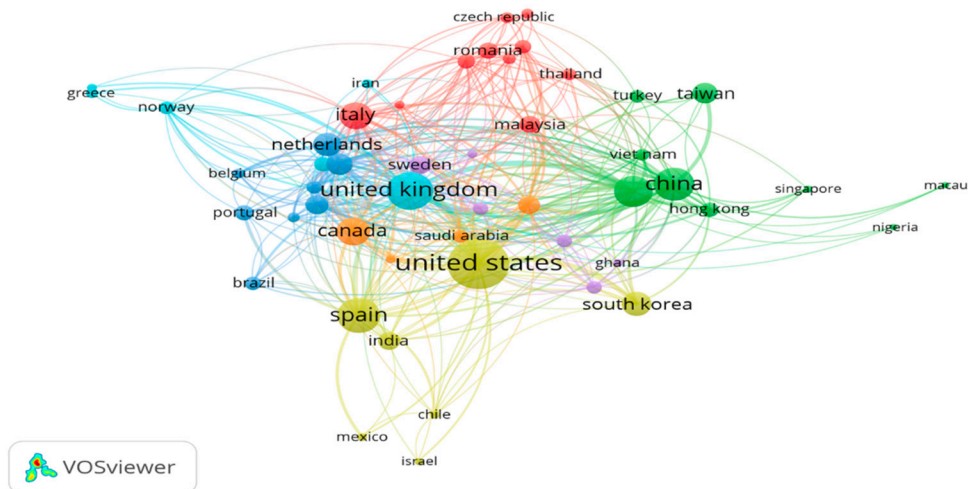

**Figure 9.** The network of cooperation considering the co-authors' locations. Source: Own elaboration.

The clusters are listed in Table 3, with these being named after the location with the largest number of articles.

**Table 3.** Locations Clusters.

| Cluster Number * | Color | Cluster Name ** | Locations |
| --- | --- | --- | --- |
| 1 | Red | Italy | Malaysia, Romania, Poland, South Africa, Thailand, Czech Republic, Indonesia, Slovakia, Lithuania |
| 2 | Green | China | Australia, Taiwan, Hong Kong, Turkey, Vietnam, Singapore, Macau, Nigeria |
| 3 | Dark blue | Netherlands | Germany, France, Portugal, Brazil, Switzerland, Austria, Belgium |
| 4 | Yellow | USA | Spain, South Korea, India, Chile, Mexico, Israel |
| 5 | Purple | Sweden | New Zealand, United Arab Emirates, Finland, Ghana, Ireland |
| 6 | Light blue | United Kingdom | Denmark, Norway, Greece, Iran, Cyprus |
| 7 | Orange | Canada | Pakistan, Saudi Arabia, Egypt |

Legend: * = see in Figure 9; ** = main location; % = network percentage. Source: Own elaboration.

Cluster 1 (red) is the largest, with 10 locations, led by Italy in association with Malaysia, Romania, Poland, and others. In cluster 2 (green), China has the greatest number of articles, in collaboration with Australia, Taiwan, Hong Kong, and others. In dark blue, cluster 3 is led by the Netherlands, which has a collaboration network with locations such as Germany, France, Portugal, and Brazil. The United States stands out in cluster 4 (yellow) in collaboration with Spain, South Korea, India, and others. Cluster 5 (purple) is headed by Sweden, collaborating with New Zealand, United Arab Emirates, Finland, Ghana, and Ireland. In cluster 6, the location publishing most is the United Kingdom, in collaboration with Denmark, Norway, Greece, Iran, and Cyprus. Finally, cluster 7 is led by Canada, collaborating with Pakistan, Saudi Arabia, and Egypt.

### 4.1.3. Keyword Analysis

Figure 10 shows the keyword network for public administration and values oriented to sustainability, based on co-occurrence. The main keywords used in the articles were "sustainability", "CSR", "sustainable development", "innovation" and "leadership". Five main groups of keywords were detected through co-occurrence analysis of the articles published on this subject.

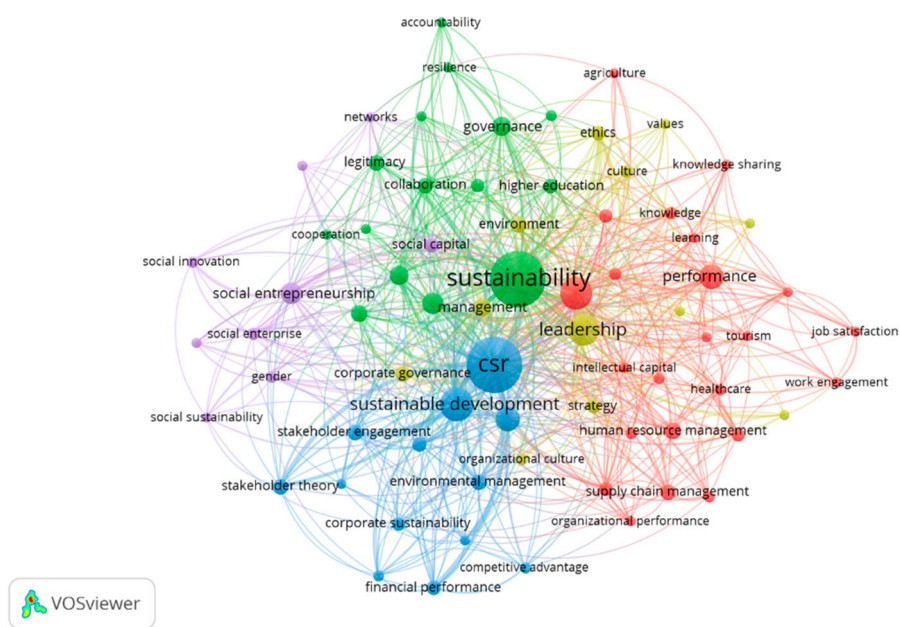

**Figure 10.** Network of keywords based on co-occurrence. Source: Own elaboration.

Cluster 1 (red) is the largest, grouping 31.94% of the keywords analyzed. The main keyword, due to its greater number of co-occurrences, is "innovation", associated with the words "performance", "human resource management", "supply chain management", "climate change", "transformational leadership", "small and medium enterprises—SMES", "entrepreneurial orientation", "knowledge", "competitiveness", "developing countries", "healthcare", "structural equation modelling", "hospitality", "intellectual capital", "job satisfaction", "organizational performance", "tourism", "agriculture", "knowledge management", "knowledge sharing", "learning", and "work engagement".

Cluster 2 (green) presents 20.83% of the keywords. The keyword with the greatest number of co-occurrences is "sustainability", associated with "entrepreneurship", "stakeholders", "governance", "social responsibility", "collaboration", "legitimacy", "higher education", "trust", "participation", "business ethics", "resilience", "accountability", "cooperation", and "firm performance". Cluster 3 (blue) groups 18.06% of the keywords analyzed. The main keyword is "CSR", associated with the words "sustainable development", "institutional theory", "environmental management", "stakeholder theory", "stakeholder engagement", "financial performance", "environmental policy", "corporate sustainability", "competitive advantage", "environmental performance", "performance measurement", and "sustainability reporting". Cluster 4 (yellow) presents 16.67% of the keywords. The word standing out is "leadership", associated with "management", "corporate governance", "environment", "ethics", "strategy", "culture", "organizational culture", "local government", "values", "implementation", and "total quality management". Finally, cluster 5 (purple) has the smallest number of keywords, presenting 12.5% of the total. The keyword with the greatest number of co-occurrences is "social entrepreneurship", associated with "social capital", "social enterprise", "social innovation", "gender", "business performance", "networks", "education", and "social sustainability". Table 4 presents the main keywords associated with the five clusters, named after the keyword with the most co-occurrences. It is worth pointing out the significant presence of organizational aspects and characteristics in the main keywords.

**Table 4.** Clusters of keywords.

| Cluster Number * | Color | Cluster Name ** | Main Keywords |
|---|---|---|---|
| 1 | Red | Innovation | Performance, human resource management, supply chain management, climate change, transformational leadership |
| 2 | Green | Sustainability | Entrepreneurship, stakeholders, governance, social responsibility, collaboration |
| 3 | Blue | CSR | Sustainable development, institutional theory, environmental management, stakeholder theory, stakeholder engagement |
| 4 | Yellow | Leadership | Management, corporate governance, ethics, strategy, culture |
| 5 | Purple | Social entrepreneurship | Social capital, social enterprise, social innovation, gender, business performance |

Legend: * = see in Figure 10; ** = main keyword. Source: Own elaboration.

These results show that innovation in the public sector is associated with both internal and external stimuli/factors [75]. In this sector, organizational factors influence the sustainability of governance and impact decision-making [76]. Moreover, CSR practice is positively associated with greater engagement by stakeholders, such as employees [77], and increased trust among service users [78]. Leadership and ethical culture are associated with better financial performance in organizations [79]. Finally, social entrepreneurship and social firms allow greater social innovation, maximizing social interests and adding social values [80].

Figure 11 presents the evolution of each keyword cluster. This diagram shows the pioneering keywords, when they first appeared, and their influence over the 28 years analyzed.

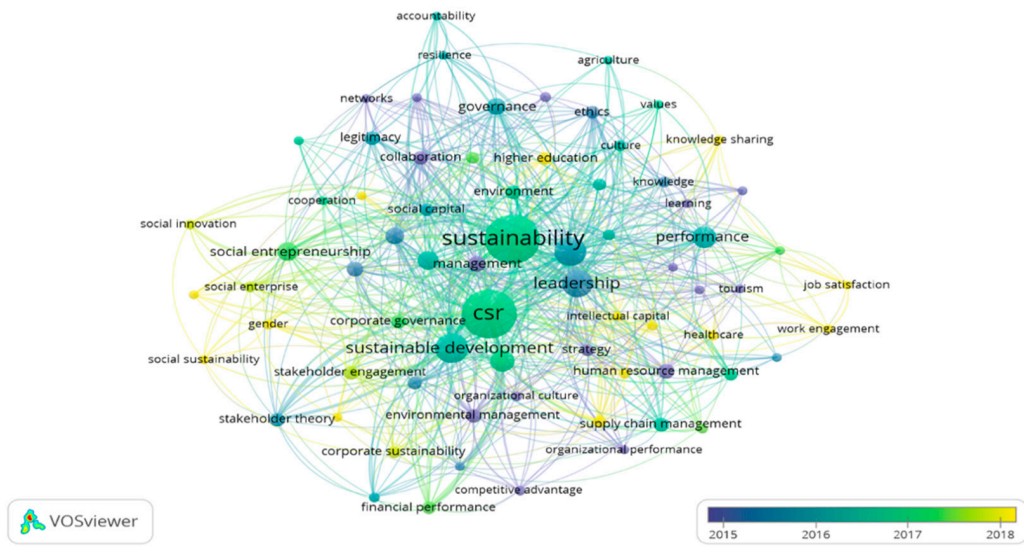

**Figure 11.** Evolution of the network of keywords based on co-occurrence. Source: Own elaboration.

The most influential keywords are seen to emerge in 2016 and 2017, where "sustainability" and "CSR" stand out. The words emerging more recently, in 2017 and 2018, show the recent interest in the social area, such as gender issues, social innovation, and social sustainability, and employee-related aspects such as engagement and work satisfaction.

*4.2. Qualitative Content Analysis*

Examination of the 2038 articles reveals a growing trend of studies on values, public administration, and sustainability (Figure 12). The last 10 years (2011–2020) represent 91.95% of all articles. In this period, 833 (40.87%) are set in the social context and address various issues, such as human resource management [81,82], organizational culture [79,83], social entrepreneurship [84,85], knowledge management [86], social learning [87], health

care [88,89], social responsibility [77], education [90–93], and others. Around 26.7% of studies address the economic dimension in this period, highlighting the topics of performance [93–95], innovation [96,97], entrepreneurship [98], competitiveness [99,100], supply chains [101–103], and others. In the last decade, the environmental dimension (24.48%) reflects topics such as water management [104], sustainable policies [105–107], adaptation to climate change [108–110], eco-tourism [111,112], environmental responsibility [113,114], sustainable development [115–117], and the circular economy [118,119].

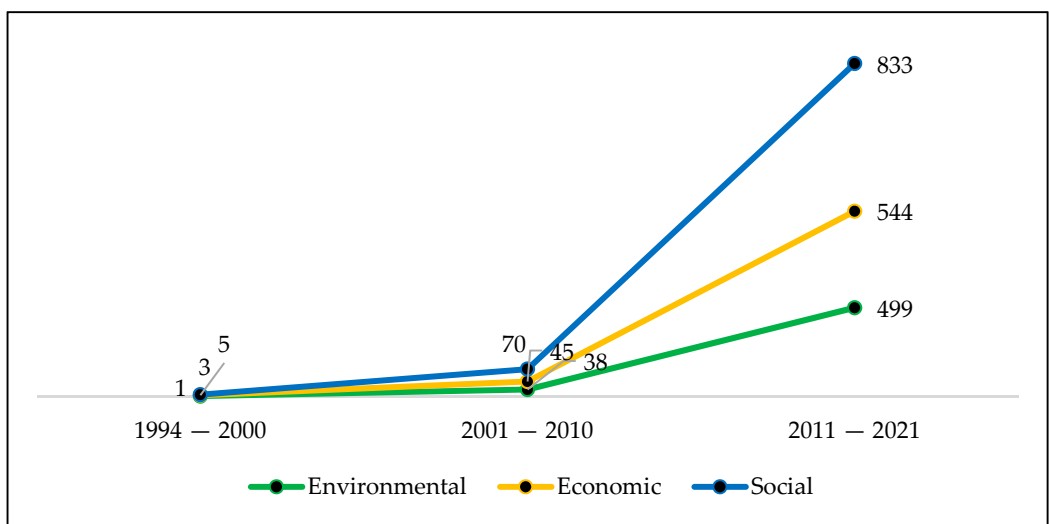

**Figure 12.** Quantity of articles distributed by decade and by type of sustainability pillar. Source: Own elaboration.

Figure 13 shows, for the total studies, the representative percentages of the types of values found in the literature review. Political values are shown in attitudes, regarding the preferences of a social group or society as a whole [120], and are most expressive, with 38.42% of studies. These are followed by cultural values (26.35%), where the individual's need to understand the values of their culture is fundamental to being able to assimilate it, become part of it, and transform it, in this way forming social learning [121]. Ethical values are defined as a set of values that guide human behavior in relation to other people in society [122], and are indicated in 18.55% of articles, followed by ecological, moral, and other values with 8.64%, 4.76%, and 3.29%, respectively.

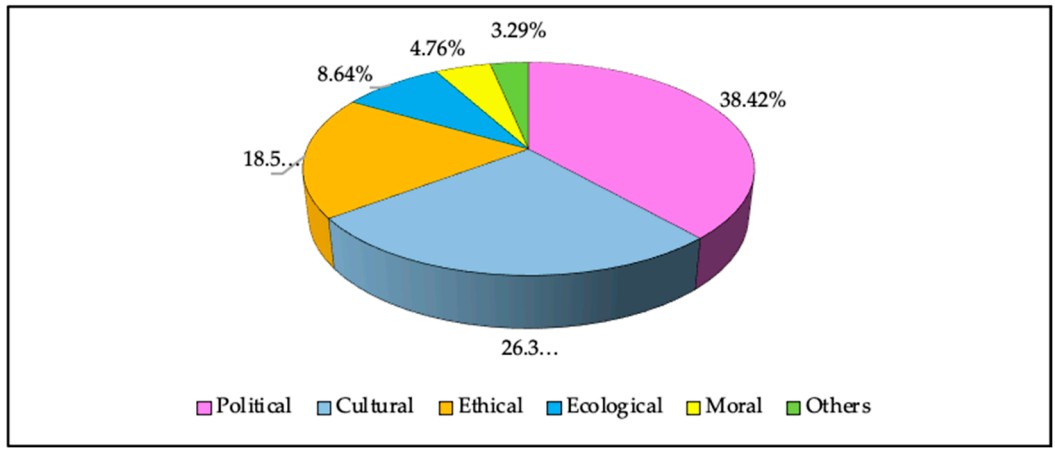

**Figure 13.** Percentage of studies by type of value. Source: Own elaboration.

The distribution of studies according to the dimensions of sustainability and the types of values is presented in Figure 14. In the social context, of the 906 articles, the

cultural value is the most expressive, with 35.43%. Political values are strongly represented in the economic and environmental dimensions with 53.37% and 39.96%, respectively. Many political decisions, whether economic, environmental, or social, are essentially choices between competing values, such as impartiality and legality, on one hand, and efficiency and effectiveness, on the other, and are seen in situations such as promoting equal opportunities. These decisions can result in conflicts between values such as efficiency, justice, equality, diversity, and merit [48].

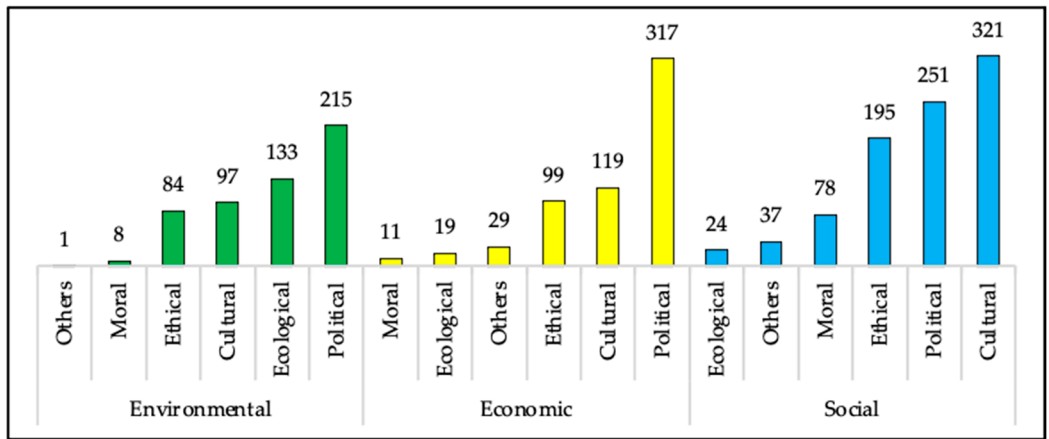

**Figure 14.** Number of studies according to sustainability pillar and type of value. Source: Own elaboration.

Concerning the major areas of public administration, Figure 15 shows the distribution of articles analyzed in this study. A relevant number of articles focus on the economy (29.34%). This is due to the relevance of the economy for sustainability, considering the organization of society and the volume and speed at which natural resources should be used [123]. Public management's participation in environmental actions has a fundamental role in ensuring a decent, sustainable future for all. In the last decade, studies addressing actions and reflections in legislating, implementing, and controlling public actions have increased (13.05%), showing their importance [124]. Sustainable social development (11.48% of studies) refers to a number of actions that aim to improve the population's quality of life, with less social inequality, assured rights, and access to services (principally education and health) giving people full access to citizenship.

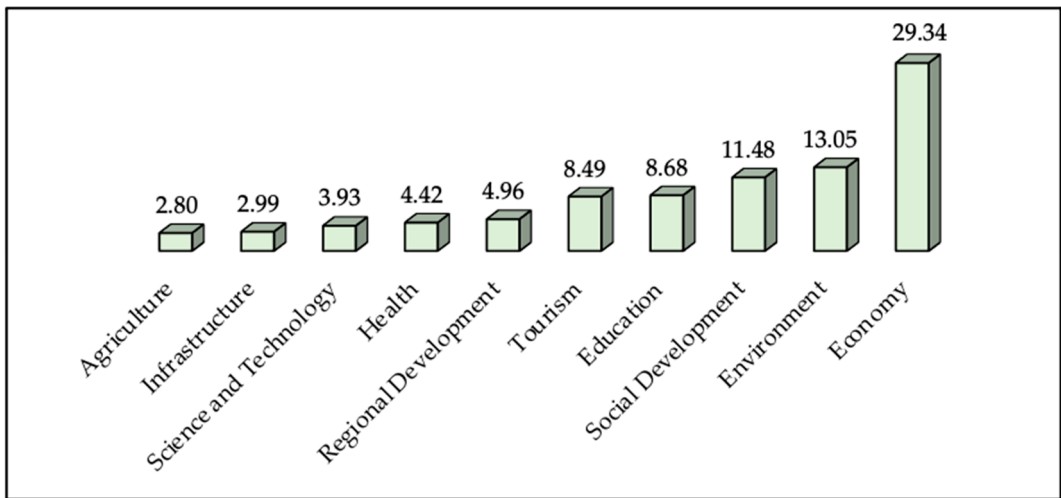

**Figure 15.** Percentage of studies distributed by area of public administration. Source: Own elaboration.

The pillars of sustainability and their relation to major areas of public administration, shown in Figure 16 and representing 71.05% of studies (1448), reveal authors' concerns regarding the economic context, as they aim to form a balance between the expanding population, social equity, and environmental conservation (343 articles). Academics are debating two different economic streams, and opinions diverge regarding the environment between the environmental or neo-classic economy, which addresses the environmental question from the point of view of pollution and natural resources, and the ecological economy, based on laws of thermodynamics and seeking to value ecological resources based on the net energy flows of ecosystems (entropy) [125,126].

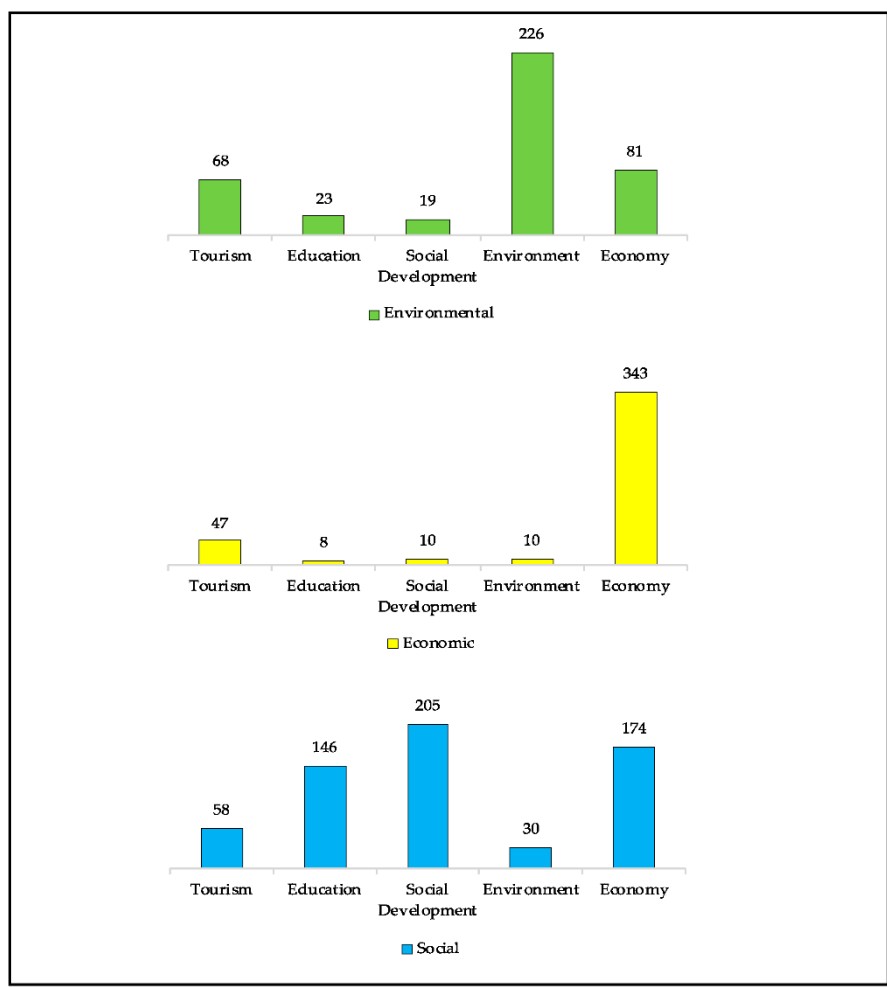

**Figure 16.** Relationship between sustainability pillars and the distribution of studies in major areas of public administration. Source: Own elaboration.

Then again, other researchers [127,128] have relevantly addressed the environmental dimension and reflections on the environment (226 articles), where the importance of natural resources for the continuity of life on Earth indicates that the economy and the environment act systemically and can be represented in the form of values, with these values being biological, ecological, and economic [129].

The relation between the social dimension and social development (205 studies) belongs to public administration's concern about social responsibility [130,131]. In the area of education, worthy of mention are studies focusing on educational policies [70,132], higher education [133,134], leadership [135], and knowledge transfer [136,137]. Research on the tourism sector and the environmental dimension has increased in the last decade and portrays authors' concern about environmentally responsible behavior [138,139], business behavior [140,141], and more.

Based on the areas of public administration that were most mentioned in this study (71.05% of all articles), their relation to the main types of values was analyzed (Figure 17). Standing out are: (i) political, cultural, and ethical values in the area of the economy; (ii) political and ecological values in the environment; (iii) political, ethical, and cultural values in social development; (iv) cultural, ethical, and political values in education; and (v) cultural, ecological, and political values in tourism.

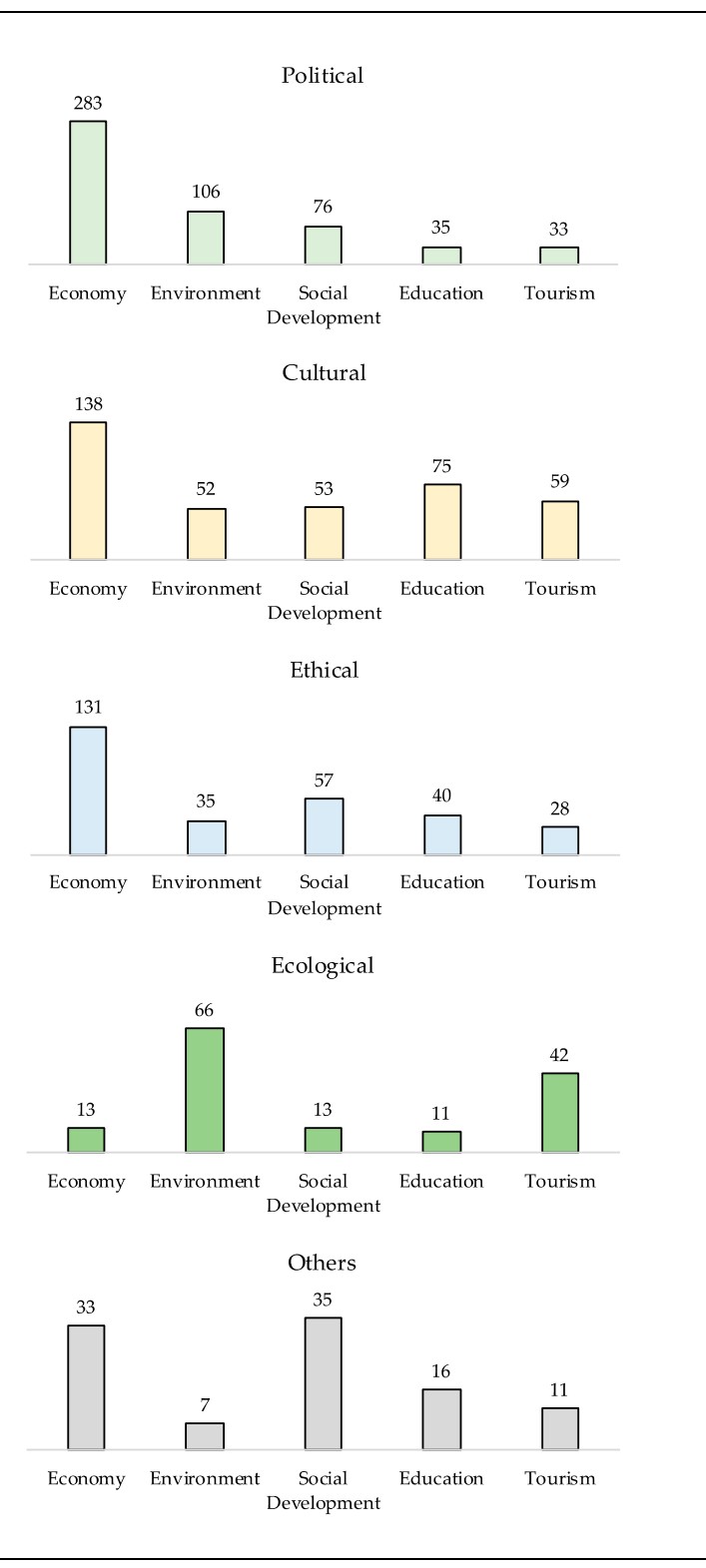

**Figure 17.** Areas of public administration and the number of studies classified according to the values. Source: Own elaboration.

Although the largest number of studies focuses on the economy, there is an undeniable growth of academic interest in environmental and social development matters. There is a strong influence of political values (533 articles), appearing in all areas of public administration and showing how important the role of public managers is in forming public policies. Also relevant are cultural and ethical values, which in fact support political values by directing good decision-making. No less importantly, ecological values have been gaining prominence among researchers, which connects to the belief in increasing human awareness concerning the conservation and sustainable use of natural resources.

The analysis of literature reviews, addressing the topic contained in the present study, means different approaches can be listed. Cowell, Downe, and Morgan [142] present the results of a survey that examined the impacts of an ethics regulation program introduced in England in 2000, which was aimed at improving the conduct of elected councilors. The ethical structure contributed to improved behavior, but the impacts were uneven across councils, reflecting broader contextual conditions—that is, managerial, political, and social.

Atkinson [143] explores the roots of environmental policy through a review and application of political literature, advocating that the government is responsible for protecting the environment in the face of rapid industrial growth. However, the inefficiency and misunderstanding of the political process, confused by a multitude of actors and interests and often inadequate resources, threaten the possibility of ensuring sustainability.

Ogunyemi and Laguda [144] develop a thematic review of the literature on ethics, governance, and sustainable practices with regard to engagement and development of the workforce. The same authors verify that existing research on ethics, governance, and sustainability in relation to workforce management can be categorized into five themes adapted from the categorization of ethical constructs in the work of Tucker et al. [145] on codes of conduct. These five themes are integrity, equality, economic efficiency and equivalence, distributive and contributory justice, and environmental concern.

The public value of e-government was the subject of a study by Twizeyimana and Andersson [146], in order to investigate the current state and what value e-government should produce. Six values, sometimes overlapping, were found: improved public services, improved administrative efficiency, open government capabilities, better ethical behavior and professionalism, greater confidence and security in the government, and better social value and well-being. These six dimensions of public value were subsequently generalized into three overarching, and also overlapping, dimensions of public value: improved public services, improved administration, and improved social value.

In relation to the current state of research on values, public administration, the public sector, and sustainability, it is worth noting that there is still limited knowledge on the adoption and coexistence of different organizational and individual values in public administration, associated with managers' level of competence, in order to foster sustainability.

## 5. Conclusions

Aiming to analyze the main trends of global research into the values attributed to sustainable public administration, in the period from 1994 to 2021, bibliometric and content analysis of 2038 articles obtained from the Scopus database was carried out. The most productive areas, authors, institutions, and locations were identified in publications on the subject of this research. The number of scientific articles per year in the period has increased significantly in the last decade, which saw the publication of a total of 1874 articles, representing 91.95% of all contributions on this issue. The most productive journals were *Sustainability Switzerland*, *Journal of Business Ethics*, and *Business Strategy and the Environment*.

The main categories identified as having the most articles published were the areas of social sciences, business management and accountability, and environmental science. As for the 10 authors with the most articles published, six are Spanish, but the top three are Chung, C.Y., from Chung-Ang University in South Korea; García-Sánchez, I.M, from the University of Salamanca in Spain; and Pérez, A., from the University of Cantabria, also in

Spain. Among the keywords associated with these authors, the most prominent are "CSR", "sustainable development", "corporate and firm value", "environmental management", "reporting", "reputation", and "stakeholders". This demonstrates a research focus on organizations' social responsibility and social development, environmental management, and the consequent influence of these on reputation and organizations' value in the eyes of their stakeholders. Analysis of co-authorship revealed that Chinese authors predominate in research collaboration networks.

The institutions with the greatest number of articles on the subject studied are the University of Queensland and Hong Kong Polytechnic University, while the main keywords associated with them are "social entrepreneurship", "non-profit organizations", "environmental management", "tourism workforce", and "CSR". Nevertheless, co-citation analysis revealed practically no research collaboration networks involving the institutions producing knowledge on this subject.

The locations contributing most to research are the United States, the United Kingdom, and Spain. The keyword analysis shows that the most relevant, due to the greater number of co-occurrences, are "sustainability", "CSR", "sustainable development", "innovation", and "leadership". Evolution over time shows a recent trend of research lines related to the social sphere, such as gender issues, social innovation, and social sustainability, and aspects related to employees, such as engagement and satisfaction at work.

As an answer to the core research question of this SLR, the typology of values adopted in the administration of public institutions appears grouped, in decreasing order of predominance, according to the components of the sustainability pillars, namely: (1) the social pillar (44.43% of studies), with cultural (1st), political (2nd), and ethical (3rd) values; (2) the economic pillar (29.13% of studies), presenting political (1st), cultural (2nd), and ethical (3rd) values; and (3) the environmental pillar (26.39% of studies), with political (1st), ecological (2nd), and cultural (3rd) values. These results reveal a predominance of political values in two of the three pillars that constitute sustainability.

The findings of this SLR contribute to filling an important gap by associating different types of organizational and individual values in the context of public administration, outlining the critical influence of managers on the sustainability of public administration. Through all the results presented, this SLR sheds light on a new descriptive and multidimensional structure, which contributes to advancing current understanding of public value, integrating different approaches, and the overlap between these values and the pillars of sustainability. For practical purposes, this multidimensional structure can be used by governments to evaluate the performance of initiatives supported by values through government policies and actions that can be used to evaluate the public value produced.

One of the limitations of this SLR is the exclusive use of the Scopus database. In future investigations, this gap could be addressed by crossing references and more comprehensive search mechanisms, including results from books, book chapters, and conference proceedings. Furthermore, the use of cluster analysis allows identification of significant concentration patterns, but this is dependent on the previous selection of search terms which, in the present SLR, had to be limited to key concepts, previously used in studies on management with public administration values, in order to ensure a dimension of the sample of selected publications that would guarantee significant results.

Although bibliometrics has some limitations because it is a form of quantitative analysis, the insertion of the qualitative analysis of the articles contributes to expanding future research trends. Concerning the future research agenda, emphasis is placed on the need to expand investigations in different areas of public administration with a focus on political and ecological values (environmental dimension). A suggestion is to carry out studies addressing the different ways in which public administration can respond to a challenging and highly uncertain political and economic context, through adopting political and cultural values (economic dimension), as well as cultural, political, and ethical values that can be adopted in the areas of education, culture, and health (among others), which can expand the scope of the areas of public administration (social dimension). In

addition, there is a need to develop studies that address perceptions about the effectiveness of policies implemented and on the evolution of sectoral systems, which can help to provide implications to ensure the sustainability of public institutions. There is also a need for research on the values attributed to sustainable public administration and studies to verify if there are differences in the impact of the three dimensions of sustainability on the reputation and value of public organizations according to different stakeholders.

Lastly, the current systematic approach of the literature not only provides an up-to-date exercise of the state of art on managing public administration institutions with values, but also opens avenues to enrich strategic orientation and governance, engaging both external and internal stakeholders, in order to build a true culture and commitment to the common good, well-being, and sustainable progress of all nations.

**Author Contributions:** Conceptualization, I.M. and J.L.; methodology, I.M. and A.C.; software, I.M. and A.C.; validation, I.M., J.L., A.C., and D.P.; formal analysis, I.M. and A.C.; investigation, I.M. and A.C.; resources, J.L.; data curation, I.M., J.L., A.C., and D.P.; writing—original draft preparation, I.M. and A.C.; writing—review and editing, J.L. and D.P.; visualization, J.L. and D.P.; supervision, J.L.; project administration, I.M.; funding acquisition, J.L. All authors have read and agreed to the published version of the manuscript.

**Funding:** This research was supported by the UBI_SantanderTotta Study Grant Programme.

**Institutional Review Board Statement:** Not applicable.

**Informed Consent Statement:** Not applicable.

**Data Availability Statement:** The data presented in this study are available on request from the corresponding author.

**Acknowledgments:** The authors acknowledge the highly valuable comments and suggestions provided by the editors and reviewers, which contributed to improvements in the clarity, focus, contribution, and scientific soundness of the current systematic literature review.

**Conflicts of Interest:** The authors declare there are no conflict of interest.

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
