# Peer review of "Public Administration and Values Oriented to Sustainability: A Systematic Approach to the Literature"

_sustainability, doi:10.3390/su13052566_

Round 1

Reviewer 1 Report

Dear authors,

first of all I would like to congratulate you on your work - this is really a fine and well-made piece of scientific work!

Still, there is some room for improvement that I would like to mention:

  • You mention in your first sentence (Line 32) the relation of values, attitudes, norms and behavior. It would be easier for the reader to follow throughout the paper if you clarified these terms and their relation among each other
  • In the introductory section I would prefer to explicitly learn about the goal and the research questions of your paper as it would be easier to navigate. At the end of the paper it would be nice to read about the answers to these research questions!
  • In your literature review it would be easier to  follow if you mentioned the names of the authors in all cases, not just in some, so one would have it easier to to compare and deduct!
  • Also the organizational values and their distinction from the personal ones are not really elaborately described, distinguished and discussed. Clarifying this would making the reading of the succeeding parts easier.
  • Your definition of sustainable development is really lapidary. SD is more than just eco and environment, but you put it in such a way here. Especially as later on you stress the social pillar, going into these aspects is really absolutely necessary, maybe also bridging the gap to the Sustainable Development Goals.
  • In line 215 you mention sustainability to be a guiding element. It can not be that - maybe you might try to rephrase this notion!
  • And your graphic I personally find not really sophisticated, especially the line where you distinguish private services and public administration. May you want to rethink it or even skip it, as it does not really add to the understanding of the paper!

As for the rest of the paper I am really happy with  your work - please bear in mind to not only mention the limitations and outlook at the end of the paper, but to also stress what were your research questions and what the answers to them!

There is some work involved in adapting some of the parts,  although I do not doubt that you will easily be able to meet with my suggestions.

Congrats again, all the best to you!

Author Response

Dear Editor-in-Chief of the Sustainability Journal, Professor Marc A. Rosen

Dear Guest Editor, Prof. Pedro Miguel Alves Ribeiro Correia

We are very pleased to have the opportunity for submitting our co-authored paper titled: Public Administration and Values Oriented to Sustainability: A Systematic Approach of Literature; to be considered in the scope of the Special Issue on: “Sustainability in Organizational Values and Public Administration”.

Firstly, we would like to thank all the reviewers for the constructive feedback and suggestions concerning the previous version of the manuscript. Secondly, we are very pleased to have had the opportunity to revise and resubmit the paper. Considering the responses to the questions raised, we provide a global overview of what was changed according to the review proposals and constructive suggestions made by the reviewers.

Yours faithfully

The Authors

Reviewer 1:

Comments and Suggestions for Authors

Dear authors,

first of all I would like to congratulate you on your work - this is really a fine and well-made piece of scientific work!

Still, there is some room for improvement that I would like to mention:

  • Q1: You mention in your first sentence (Line 32) the relation of values, attitudes, norms and behaviour. It would be easier for the reader to follow throughout the paper if you clarified these terms and their relation among each other.
  • A1: We acknowledge the reviewer’s comment. For addressing it, the following sentence clarifying the relationships among values, norms, and behaviour, was inserted in the Introductory item:

Introduction, Line 32: The main motivation of this systematic literature review is related to the scarcity of literature reviews that deal with the problem of public administration and values oriented towards sustainability. In this vein, it is necessary to map political, cultural, ethical, moral, aesthetic, ecological, vital, spiritual and religious values to understand the different ways in which the organizational and individual values have been addressed in the literature, as well as the contributions of skills managerial techniques and moral competence have a behavioral effect, which allows achieving a balanced exercise of sustainability. In this sense, the administration of public institutions requires the adoption of ethical principles and values oriented towards the effective and efficient use of public resources with the ultimate aim of contributing to the increase of social well-being.

  • Q2: In the introductory section I would prefer to explicitly learn about the goal and the research questions of your paper as it would be easier to navigate. At the end of the paper it would be nice to read about the answers to these research questions!

A2: We acknowledge the reviewer’s comment. For addressing it was included in the concluding remarks item a sentence stating the answer to the core research question, in the following terms:

Conclusion, Line 715: As an answer to the core research question of this SLR, the typology of values ​​adopted in the administration of public institutions appears grouped, in decreasing order of predominance, according to the components of the sustainability pillars, namely: (1) the social pillar (44.43% of the studies), with cultural (1st), political (2nd) and ethical (3rd) values; (2) the economic pillar (29.13% of the studies) presenting the political (1st), cultural (2nd) and ethical (3rd) values; ​​and (3) the environmental pillar (26.39% of the studies), the political (1st), ecological values (2nd) and cultural (3rd). These results reveal a predominance of political values ​​in two of the three pillars that constitute sustainability.

  • Q3: In your literature review it would be easier to follow if you mentioned the names of the authors in all cases, not just in some, so one would have it easier to compare and deduct!

A3: We appreciate the reviewer's comment, with which we agree, however, given that the formatting rules of the Sustainability Journal point to the use of sequential numbering of references and allows mentioning only a few authors, we chose to keep the mention of some authors, throughout the text, in order to prevent an excessive increase of the total number of pages of the revised version of the paper.

  • Q4: Also the organizational values and their distinction from the personal ones are not really elaborately described, distinguished and discussed. Clarifying this would making the reading of the succeeding parts easier.

A4: We acknowledge the reviewer’s comment, which was addressed by elaborating further the distinction between organizational values and personal values in the following paragraphs:

Item 2.2, Line 149: Renshaw, Parry & Dickmann [28] point out that many studies address the term: organizational values; without a proper definition, probably due to the difficulty of conceptualizing it, bearing in mind that values are often positioned as constructs at the individual level [29]. However, it should be noted that organizational values are important components of the organizational culture [30] and principles that are responsible for the successful management of organizations [31]. As concerns to individual values, these can be defined as the internalized beliefs in individuals about the way they should behave [32], according to their personal experiences [33], culture and social system where they are inserted [31,33].

Thus, it is also important to note that there is a tension between both types of values. Public employees are exposed to conflicting demands, which they must meet [34]. This conflict stems from the need to respond to citizens, align their decisions with the interests of co-workers [35] and adapt their preferences, functions and identities to the organization in which they are located. At the same time, public institutions need to meet the demands, presenting the best results, using fewer resources and have the challenge of integrating these employees into the organization and its standards. This adjustment process between the parties corresponds to the so called organizational socialization [36].

Individuals who are part of public organizations are often frustrated with their restrictions and ambiguities [37] and with the organizational ownership of values in the public service [38]. Considering the importance of congruence between organizational and individual values, in terms of the attitudes and behaviors of individuals [39,40], for reaching a better individual performance [39,40,41] and for ensuring the organization' success [41], the key-role played by the leaders to encourage this alignment of values is considered fundamental. In this ambit, transformational leadership is a powerful a tool that can be used by managers to promote an oriented process for alignment and congruence of values in the public service [41,42].

  • Q5: Your definition of sustainable development is really lapidary. SD is more than just eco and environment, but you put it in such a way here. Especially as later on you stress the social pillar, going into these aspects is really absolutely necessary, maybe also bridging the gap to the Sustainable Development Goals.

A5: We acknowledge the reviewer’s comment, which led us to expand the definition of sustainable development in connection with the SDGs, by introducing the sentences made available below:

Item 2.3, Line 210: Following the UNESCO (United Nations Educational, Scientific and Cultural Organization’s vision, sustainable development is the comprehensive paradigm of the United Nations, described by the 1987 Bruntland Commission Report as the “development that meets the needs of the present without compromising the ability of future generations to meet their own needs” [52]. Sustainable development has four interconnected dimensions (e.g. society, environment, culture and economy) and sustainability is a paradigm facing the future in which environmental, social and economic considerations are balanced in the search for improving the quality of life, being considered a long-term goal, while sustainable development refers to the various processes and paths to achieve it. In 2015, all Member States of the United Nations adopted the Sustainable Development Goals containing 169 goals that countries seek to achieve by 2030. According to Clar et a. [53], the public sector faces some barriers to the adoption of policies that lead to sustainable development, such as lack of commitment, inadequate or unclear responsibilities, inadequate cooperation between political actors, insufficient financial and human resources, lack of evidence or certainty in relation to global scenarios [53]. Promoting governance and providing better ways to link science to policymaking enables decisions to be based on good research that emphasizes trade-offs and multiple possibilities for action [54].

  • Q6: In line 215 you mention sustainability to be a guiding element. It can’t be that - maybe you might try to rephrase this notion!

A6: We acknowledge the reviewer’s comment, which was addressed by rephrasing the following sentences:

Item 2.3, Line 263: In this study, both organizational and individual values are connected with technical skills at the level of competence of the managers who have a critical influence on the sustainability of public institutions. The combination of values, skills, and level of competence, can stimulate new entrepreneurial practices that will lead to a sustainable pathway.

  • Q7: And your graphic I personally find not really sophisticated, especially the line where you distinguish private services and public administration. May you want to rethink it or even skip it, as it does not really add to the understanding of the paper!

A7: We acknowledge the reviewer’s comment. Accordingly, Figure 1 has been changed.

  • Q8: As for the rest of the paper I am really happy with your work - please bear in mind to not only mention the limitations and outlook at the end of the paper, but to also stress what were your research questions and what the answers to them!

A8: We acknowledge the positive incentives and comments provided by the reviewer. Thus, in the Conclusions item, aside from reinforcing the answers provided to the core research question (as previously presented in A2), the limitations of the analysis were presented in a more extensive way, adding the following sentences and argumentation:

Conclusions, Line 734: One of the limitations of this SLR is the exclusive use of the Scopus database. In future investigations, this gap could be addressed by crossing references and more comprehensive search mechanisms, including results from books, book chapters and conference proceedings. Furthermore, the use of cluster analysis allows the identification of significant concentration patterns, however, it is dependent on the previous selection of search terms that, in the present SLR, had to be limited to key concepts, previously used in studies on management with public administration values, in order to ensure a dimension of the sample of selected publications that would guarantee significant results. Although bibliometrics has some limitations because it is a quantitative analysis, the insertion of the qualitative analysis of the articles contributes to expanding future research trends.

Reviewer 2 Report

The Introduction section state the motivation for the work presented in your paper and prepare readers for the structure of the paper. I cannot see which is the object. It is important because shows the specific outcomes you expect to achieve through your research and that will enable you to meet the aim of your research. Please include it in the line 76.

Line 57: Why did the authors decide to use the Scopus database? What are its advantages over the others? And how are the limitations?

DISCUSSION: A more critical position is required. The literature review should be extended. More comments about the results and comparisons with similar studies from literature are required.

Given that the discussion of results in a scientific article shows the real contribution to knowledge, I recommend that the results be interpreted with more details and that the implications of these results be reflected upon. Discuss your results in order of most to least important; Compare your results with those from other studies; Discuss what your results may mean for researchers in the same field as you, researchers in other fields, and the general public. How could your findings be applied? State how your results extend the findings of previous studies.

The conclusions present a very superficial view. I do recommend authors improve this section.

Author Response

Dear Editor-in-Chief of the Sustainability Journal, Professor Marc A. Rosen

Dear Guest Editor, Prof. Pedro Miguel Alves Ribeiro Correia

We are very pleased to have the opportunity for submitting our co-authored paper titled: Public Administration and Values Oriented to Sustainability: A Systematic Approach of Literature; to be considered in the scope of the Special Issue on: “Sustainability in Organizational Values and Public Administration”.

Firstly, we would like to thank all the reviewers for the constructive feedback and suggestions concerning the previous version of the manuscript. Secondly, we are very pleased to have had the opportunity to revise and resubmit the paper. Considering the responses to the questions raised, we provide a global overview of what was changed according to the review proposals and constructive suggestions made by the reviewers.

Yours faithfully

The Authors

Reviewer 2:

Comments and Suggestions for Authors

  • Q1: The Introduction section state the motivation for the work presented in your paper and prepare readers for the structure of the paper. I cannot see which is the object. It is important because shows the specific outcomes you expect to achieve through your research and that will enable you to meet the aim of your research. Please include it in the line 76.

A1: We acknowledge the reviewer’s comment, which is addressed in the introductory item by presenting the object of the current systematic literature review, including the following sentence:

Introduction, Line 82: In this context of analysis, it is considered relevant to verify the typology of the values adopted in the administration of public institutions that combined with the technical skills and the moral competence of the managers, allow to ensure the ultimate goal of sustainability.

  • Q2: Line 57: Why did the authors decide to use the Scopus database? What are its advantages over the others? And how are the limitations?

A2: We acknowledge the reviewer’s comment. For addressing it, the following justification for using the Scopus database was added to the manuscript:

Added (Line 280), in item 3 Materials and Methods: The Scopus database was chosen due to its multidisciplinary nature and large coverage. Adding to the previous, it is peer-reviewed, has daily updates, and has resources that assist the user in the searches carried out on the website and creation lists for storing documents during the search session in the database, with structured searches by author and subject. As main advantages, it should be outlined: (i) inclusion of titles made available in Open Access; (ii) wide coverage in terms of science and technology magazines; (iii) tools for identifying authors; (iv) automatic generation of the h-index; (v) inclusion of more European content than Web of Science (WoS); and (vi) integration of more languages than English. An interesting feature to be outlined, is that, although, the Scopus database was not designed as a citation index, it includes citations from articles since 1996.

  • Q3: DISCUSSION: A more critical position is required. The literature review should be extended. More comments about the results and comparisons with similar studies from literature are required.

Given that the discussion of results in a scientific article shows the real contribution to knowledge, I recommend that the results be interpreted with more details and that the implications of these results be reflected upon. Discuss your results in order of most to least important; Compare your results with those from other studies; Discuss what your results may mean for researchers in the same field as you, researchers in other fields, and the general public. How could your findings be applied? State how your results extend the findings of previous studies.

A3: We acknowledge the reviewer’s comment, which was addressed by contrasting the current findings with previous perspectives, as well as extending the findings of previous studies, through the inclusion of the following paragraphs:

Item 4.2. Qualitative Content Analysis, Line 649: The analysis of literature reviews, which address the topic contained in the present study, allows to list different approaches. Cowell, Downe and Morgan [145] present the results of a survey that examined the impacts of an ethics regulation program introduced in England in 2000, aiming to improve the conduct of elected councilors, where the ethical structure contributed to the improved behavior, but the impacts were uneven across councils, reflecting broader contextual conditions, that is, managerial, political and social.

Atkinson [146] explore the roots of environmental policy through a review and application of political literature, advocating that the government is responsible for protecting the environment in the face of rapid industrial growth. Albeit the inefficiency and misunderstanding of the political process, confused by a multitude of actors and interests and, often, inadequate resources, threaten the possibility of ensuring sustainability.

Ogunyemi and Laguda [147] develop a thematic review of the literature on ethics, governance, and sustainable practices with regard to engagement and development of the workforce. The same authors verify that existing research on ethics, governance and sustainability in relation to workforce management can be categorized into five themes adapted from the categorization of ethical constructs in the work of Tucker et al. [148] on codes of conduct. These five themes are integrity, equality, economic efficiency and equivalence, distributive and contributory justice and environmental concern.

The public value of e-government was the subject of a study by Twizeyimana and Andersson [149], in order to investigate the current state and what value e-government should produce. Six values, sometimes overlapping, were found: improved public services; improving administrative efficiency; open government capabilities; better ethical behavior and professionalism; greater confidence and security in the government; and better social value and well-being. These six dimensions of public value were subsequently generalized into three overarching, and also overlapping, dimensions in public value of Improved Public Services, Improved Administration and Improved Social Value.

In relation to the current state of research on values, public administration, public sector and sustainability, it is worthwhile to note that there is still limited knowledge on the adoption and coexistence of different organizational and individual values in public administration, associated with the level of competence of the managers, in order to foster sustainability.

  • Q4: The conclusions present a very superficial view. I do recommend authors improve this section.

A4: We acknowledge the reviewer’s comment, which was addressed by improving the presentation of the contributions of the current systematic literature review, as well as potential applications, in the following sentences:

Conclusion, Line 724: The findings found in the current SLR contributes to the filling of an important gap, by associating different types of values, organizational and individual, in the context of public administration, outlining the critical influence that managers have on the sustainability of public administration. Through all the results presented, this SLR sheds light on a new descriptive and multidimensional structure, which contribute to the advancement of the current understanding of the public value, integrating different approaches, and on the overlap between these values and the pillars of sustainability. For practical purposes, this multidimensional structure can be used by governments to evaluate the performance of initiatives supported by values through government policies and actions that can be used for evaluating the public value produced.

Reviewer 3 Report

This work is nicely presented and gives an important contribution to the field. It is based on a systematic review and bibliometric analysis of the current state of the art (1994-2020) on the field of Public Administration and Values Oriented to Sustainability.

The introductions sections is very coherent and the systematic review is very well conducted. It would be nice to have a justification for the timeline.

Also, in line 37 when the authors stated "These authors" their identification should be provided.

Until figure 13 the quality is good but after figure 14 would be nice to have those improved.

I believe that table 5 should be transformed in text once in conclusion section tables are not very usual.

Author Response

Dear Editor-in-Chief of the Sustainability Journal, Professor Marc A. Rosen

Dear Guest Editor, Prof. Pedro Miguel Alves Ribeiro Correia

We are very pleased to have the opportunity for submitting our co-authored paper titled: Public Administration and Values Oriented to Sustainability: A Systematic Approach of Literature; to be considered in the scope of the Special Issue on: “Sustainability in Organizational Values and Public Administration”.

Firstly, we would like to thank all the reviewers for the constructive feedback and suggestions concerning the previous version of the manuscript. Secondly, we are very pleased to have had the opportunity to revise and resubmit the paper. Considering the responses to the questions raised, we provide a global overview of what was changed according to the review proposals and constructive suggestions made by the reviewers.

Yours faithfully

The Authors

Reviewer 3:

Comments and Suggestions for Authors

This work is nicely presented and gives an important contribution to the field. It is based on a systematic review and bibliometric analysis of the current state of the art (1994-2020) on the field of Public Administration and Values Oriented to Sustainability.

  • Q1: The introductions sections is very coherent and the systematic review is very well conducted. It would be nice to have a justification for the timeline.

A1: We acknowledge the reviewer’s comment. Thus, a justification for the timeline was added in the following terms:

Including the justification for the timeline used, in item 4.1.1 Publications, citations and research areas (Line 324): Figure 3 shows the trend of evolution of publications on the topic studied. From the final selection (n = 2,038), the articles were stratified according to the dates of publication, which cover the 1994-2020 period. From the total sum of 2,038 articles identified, 1,449 were published in the last 5 years, 2016 to 2020, that is, 71.09% of all scientific production in these 28 years. This result reveals the growing interest and relevance of the topic.

  • Q2: Also, in line 37 when the authors stated "These authors" their identification should be provided.

A2: We acknowledge the reviewer’s comment, which led us to the following amendment: Introduction, authors added (Line 45): “For the same authors [1], having values represents understanding the importance of ethical processes for decision-making; being ethical helps in choosing the correct values.”

  • Q3: Until figure 13 the quality is good but after figure 14 would be nice to have those improved.

A3: We acknowledge the reviewer’s comment. Accordingly, Figures 14 to 17 have been reformulated.

  • Q4: I believe that table 5 should be transformed in text once in conclusion section tables are not very usual.

A4: We acknowledge the reviewer’s comment, which was incorporated by making available the table's contents in the following sentences:

Conclusion, Line 742: Although bibliometrics has some limitations because it is a quantitative analysis, the insertion of the qualitative analysis of the articles contributes to expanding future research trends. Concerning the future research agenda, the emphasis is placed on the need to expand the investigations in different areas of public administration with a focus on political and ecological values ​​(Environmental dimension); it is suggested to pursue studies that address the different ways in which public administration can respond to a challenging and highly uncertain political and economic context, through the adoption political and cultural values ​​(Economic dimension), as well as cultural, political and ethical values ​​that can be adopted in the areas of Education, Culture, Health (among others), contributing to expand the scope of the areas of public administration (Social dimension). In addition, there is a need to develop studies that address perceptions about the effectiveness of policies implemented and on the evolution of sectoral systems, which can help to provide implications for ensuring the sustainability of public institutions, as well as research on the values ​​attributed to sustainable public administration and studies to verify if there are differences in the impact of the three dimensions of sustainability, on the reputation and value of public organizations, according to different stakeholders.

Round 2

Reviewer 1 Report

Dear authors,

thank you for attending to almost all of my suggestions,  I would only suggest some proof-reading - there still seem to have remained some spelling mistakes.

Otherwise congrats, very well done!

Kind regards from Austria,

Ulli Gelbmann

Author Response

Dear Editor-in-Chief of the Sustainability Journal, Professor Marc A. Rosen

Dear Guest Editor, Prof. Pedro Miguel Alves Ribeiro Correia

We are very pleased to have the opportunity for submitting our co-authored paper titled: Public Administration and Values Oriented to Sustainability: A Systematic Approach of Literature; to be considered in the scope of the Special Issue on: “Sustainability in Organizational Values and Public Administration”.

Firstly, we would like to thank all the reviewers for the constructive feedback and suggestions concerning the previous version of the manuscript. Secondly, we are very pleased to have had the opportunity to revise and resubmit the paper. Considering the response to the question raised, we provide a global overview of what was changed according to the review proposal and constructive suggestion made by the reviewer.

Yours faithfully

The Authors

Reviewer 1:

Comments and Suggestions for Authors

Dear authors,

thank you for attending to almost all of my suggestions, I would only suggest some proof-reading - there still seem to have remained some spelling mistakes.

Otherwise congrats, very well done!

  • A1: We acknowledge the reviewer’s comment, as well as the positive incentives. For addressing this comment, the manuscript was proofread by an English native. All the changes are blue coloured.